# Identification of EGFR-Related LINC00460/mir-338-3p/MCM4 Regulatory Axis as Diagnostic and Prognostic Biomarker of Lung Adenocarcinoma Based on Comprehensive Bioinformatics Analysis and Experimental Validation

**DOI:** 10.3390/cancers14205073

**Published:** 2022-10-17

**Authors:** Mingxi Jia, Shanshan Feng, Fengxi Cao, Jing Deng, Wen Li, Wangyan Zhou, Xiang Liu, Weidong Bai

**Affiliations:** 1National Engineering Laboratory for Deep Process of Rice and Byproducts, College of Food Science and Engineering, Central South University of Forestry and Technology, Changsha 410004, China; 2Collaborative Innovation Center for Modern Grain Circulation and Safety, College of Food Science and Engineering, Nanjing University of Finance and Economics, Nanjing 210023, China; 3College of Light Industry and Food Sciences, Zhongkai University of Agriculture and Engineering, Guangzhou 510225, China; 4College of Life Sciences and Chemistry, Hunan University of Technology, Zhuzhou 412007, China; 5Second Affiliated Hospital of Luohe Medical College, Luohe Medical College, Luohe 462000, China; 6Department of Medical Record, Hengyang Medical School, The First Affiliated Hospital, University of South China, Hengyang 421001, China; 7Department of Thoracic Surgery, Hengyang Medical School, The Second Affiliated Hospital, University of South China, Hengyang 421001, China

**Keywords:** lung adenocarcinoma, ceRNA network, LINC00460/MCM4 axis, drug targets, prognosis

## Abstract

**Simple Summary:**

While the epidermal growth factor receptor (EGFR) is an important target for lung adenocarcinoma (LUAD) therapy, acquired resistance is still inevitable. A comprehensive bioinformatics analysis strongly suggested that the closely related LINC00460-mir-338-3p-MCM4 ceRNA network of EGFR plays an important role in the diagnosis and prognosis of LUAD. High expression of LINC00460 and MCM4 predicts shorter patient survival. Univariate and multivariate regression analyses demonstrated that higher expression of LINC00460 and MCM4 was significantly associated with tumor size, lymph node metastasis, distant metastasis and TNM stage. A multi-gene regulation model analysis revealed that the LINC00460 (downregulation)—mir-338-3p (upregulation))—MCM4 (downregulation) pattern significantly improved the overall survival of LUAD patients. We also verified the expression of these genes in LUAD cell lines and tumor tissues by RT-PCR and immunohistochemistry. Finally, the possible targeted drugs of MCM4 were queried through the drug database platform, hoping to solve its drug resistance problem by targeting EGFR-related genes.

**Abstract:**

**Background:** Lung adenocarcinoma (LUAD) is one of the most aggressive and lethal tumor types and requires effective diagnostic and therapeutic targets. Though the epidermal growth factor receptor (EGFR) is an important target for LUAD therapy, acquired resistance is still inevitable. In recent years, the regulation of the EGFR by competing endogenous RNAs (ceRNAs) has been extensively studied and significant progress has been made. Therefore, we aim to find new targets for the diagnosis and treatment of LUAD by analyzing the EGFR-related ceRNA network in LUAD and expect to address the problem of EGFR resistance. **Methods:** We identified differentially expressed lncRNAs, miRNAs and mRNAs closely associated with the EGFR by analyzing transcriptome data from LUAD samples. Comprehensive bioinformatics analysis strongly suggests that the LINC00460—mir-338-3p—MCM4 ceRNA network plays an important role in the diagnosis and prognosis of LUAD. The effects of different patterns of the LINC00460/MCM4 axis on the overall survival of patients with LUAD were analyzed by a polygene regulation model. We also verified the expression of these genes in LUAD cell lines and tumor tissues by RT-PCR and immunohistochemistry. The functional enrichment analysis and targeted drug prediction of the MCM4 gene were performed. **Results:** Survival analysis indicated that high expressions of LINC00460 and MCM4 predict a shorter survival period for patients. Univariate and multivariate regression analyses demonstrated that higher expressions of LINC00460 and MCM4 were significantly associated with tumor size, lymph node metastasis, distant metastasis and TNM stage. A multi-gene regulation model analysis revealed that the LINC00460 (downregulation)—mir-338-3p (upregulation)—MCM4 (downregulation) pattern significantly improved the overall survival of LUAD patients (*p* = 0.0093). RT-PCR and immunohistochemical experiments confirmed our analytical results. In addition, the functional enrichment analysis indicated that MCM4-related genes were mainly enriched in the cell cycle and cell division. A functional association network analysis showed that MCM4 was closely related to the EGFR. Finally, the possible targeted drugs of MCM4 were queried through the drug database platform, hoping to solve its drug resistance problem by targeting EGFR-related genes. **Conclusions:** In summary, the LINC00460/MCM4 axis can be used as a potential new perspective for targeting EGFR genes in precision medicine and is expected to serve as a diagnostic, prognostic and drug target for LUAD.

## 1. Introduction

Lung cancer is the most common and deadliest form of cancer. Non-small cell lung cancer (NSCLC) accounts for approximately 80% of all lung cancers and has a low five-year survival rate [1,2]. LUAD is the most common histological subtype of NSCLC, and the five-year survival rate of patients represents only 15% of them [3]. Therefore, it is still very important to determine new diagnostic, prognostic and drug resistance biomarkers and treatment targets for LUAD research. 

In tumor cells, epidermal growth factor receptor (EGFR) activity may be dysregulated due to mutations, increased gene copy number or protein overexpression [4,5]. Among 441 LUAD patients, 218 (49.4%) patients had wild-type EGFR, and 223 (50.6%) had mutant EGFR [6]. Similar studies have shown that approximately 60% of NSCLC cases associated with poor prognosis have EGFR overexpression or constitutive activation [7]. Overexpression or mutation of the EGFR gene greatly promotes cell growth and division in NSCLC. Several pieces of research have indicated that the overexpression of EGFR is related to the low survival rate, frequent lymph node metastasis and poor chemotherapy sensitivity of NSCLC patients [7,8]. Furthermore, most EGFR mutations in NSCLC occur in the exons of the receptor tyrosine kinase domain [9]. Although EGFR tyrosine kinase inhibitors (EGFR-TKIs), including erlotinib and gefitinib, have shown initial efficacy in 30% of NSCLC patients with EGFR mutations in the past few decades, secondary resistance often occurs in EGFR-TKIs treatment [10,11]. Currently, multiple mechanisms of secondary resistance to EGFR-TKIs, including primary or secondary T790M point mutations, human epidermal growth factor receptor 2 (HER2) amplification, mesenchymal epithelial cell transforming factor (MET) amplification or activation of bypass signaling pathways by phosphatidylinositol 3 kinase (PI3K) mutations and epithelial –mesenchymal transitions (EMT) have been clarified. However, the combination of EGFR-TKIs with platinum or other cytotoxic chemotherapeutic agents did not achieve the expected prolongation of survival in NSCLC patients, resulting in increased toxicity and side effects [12,13]. Therefore, it is of great significance to deeply study the molecular mechanism of EGFR for the diagnosis of NSCLC, the development of new drugs and the formulation of new treatment strategies to prolong the survival of patients.

In recent years, the regulation of EGFR by competing endogenous RNAs (ceRNAs) has been extensively studied [14,15]. Long non-coding RNA (lncRNA) can regulate gene expression at multiple levels through epigenetic regulation, transcription regulation and post-transcriptional regulation, and then participate in a variety of biological processes [16,17,18]. In the lncRNA-miRNA-mRNA ceRNA network, lncRNA can regulate mRNA expression through sponge adsorption of miRNA [19,20,21]. Studies have shown that EGFR, mRNA and protein levels are regulated by a large number of protein-coding and noncoding RNAs, most of which are mRNAs unrelated to EGFR function, but capable of “protecting” it from “attack” by their shared miRNAs [22,23]. For example, the lncRNA SNHG16 is highly expressed in gliomas and acts as a ceRNA to regulate EGFR by sponging miR-373-3p to activate the PI3K/AKT pathway, thereby exerting oncogenic effects [24]. These EGFR-centered ceRNA regulatory networks are very important. Several studies have shown that high expression of EGFR is an important indicator of poor prognosis in LUAD [25]. High levels of EGFR expression are associated with resistance to chemotherapeutic drugs. A high expression of EGFR was associated with increased gene copy numbers, and non-small cell lung cancer patients given gefitinib were associated with significantly improved responses, lower progression rates and improved survival compared with tumors with low EGFR expression [26].

Based on the above description, we aim to find new targets for the diagnosis and treatment of LUAD by analyzing the EGFR-related ceRNA network. We conducted a systematic and comprehensive study on the RNAseq and miRNAseq data of LUAD from TCGA. A ceRNA network closely related to the EGFR was identified by expression analysis and survival analysis. The correlation between different expression patterns and overall survival (OS) was analyzed using a multi-gene regulation model. We also verified the expression of these genes in LUAD cell lines and tumor tissues by RT-PCR and immunohistochemistry. Finally, the possible targeted drugs of the target gene were queried through the drug database platform, hoping to solve its drug resistance problem by targeting EGFR-related genes.

## 2. Materials and Methods

### 2.1. Data Preparation and Processing

The RNA data and clinical information of lung adenocarcinoma samples were obtained from the TCGA database (https://portal.gdc.cancer.gov/, accessed on 18 January 2021). Finally, 515 tumor samples and 57 paracancerous samples were identified as research objects by matching the patient numbers corresponding to the lncRNA, mRNA and miRNA samples. 

### 2.2. Identify Different Expression RNAs in LUAD

According to the median expressed in the EGFR (median value = 6616), 515 LUAD patients were divided into EGFR^low^ group (EGFR < 6616, *n* = 257) and EGFR^high^ group (EGFR ≥ 6616, *n* = 258). Performing the differential expression analysis in EGFR^low^ and EGFR^high^ LUAD samples. The differentially expressed lncRNAs had thresholds of |log_2_FC| > 0.70 and *p* < 0.05, the differentially expressed miRNAs and mRNAs had thresholds of |log2FC| > 0.50 and *p* < 0.05. 

### 2.3. Construction of ceRNA Network

The ceRNA network was constructed according to the method described previously [27]. Briefly: (1) Prediction of potential miRNAs for DElncRNA targeting through the mircode database (http://www.mircode.org/, accessed on 20 March 2022); (2) miRDB databases (http://www.mirdb.org/miRDB/, accessed on 20 March 2022) and the Targetscan database (http://www.targetscan.org/, accessed on 20 March 2022) were used to identify the downstream targets (mRNAs) of miRNAs; (3) The lncRNA-miRNA-mRNA ceRNA network was visualized by Cytoscape v3.7.0.

### 2.4. Survival Analysis

A Kaplan–Meier survival analysis displayed the correlation between the expression of DERNAs in the ceRNA network and the prognostic survival of LUAD patients. The survival status and time of LUAD patients were obtained from TCGA clinical data. Associations between candidate genes and overall survival (OS) were analyzed to determine biological prognostic markers by univariate and multivariate Cox regression. In addition, a novel analysis model analyzed the correlation between different expression combination patterns of three genes (LINC00460, mir-338-3p and MCM4). LINC00460 was divided into high expression group “L^+^” and low expression “L^−^” based on their median expression values. Similarly, mir-338-3p was classified into “m^+^” and “m^−^” groups, and MCM4 was divided into “M^+^” and “M^−^” groups. Eight combined expression patterns were ultimately obtained, including: L^+^/m^+^/M^+^, L^+^/m^+^/M^−^, L^+^/m^−^/M^−^, L^−^/m^+^/M^−^, L^−^/m^+^/M^+^, L^+^/m^−^/M^+^, L^−^/m^−^/M^−^ and L^−^/m^−^/M^+^. The regulatory mechanism of the LINC00460—mir-338-3p—MCM4 axis on the prognosis and survival of LUAD patients was analyzed through a multi-gene interaction model. 

### 2.5. Total RNA Extraction and Quantitative RT-PCR

Lung adenocarcinoma cell lines (A549, PC-9 and H1299) and the normal bronchial epithelial cell line (BEAS-2B) were cultured in 1640 medium (Thermo Fisher Scientific, Waltham, MA, USA) containing 10% fetal bovine serum (FBS) (Thermo Fisher Scientific, Waltham, MA, USA), 100 U/mL streptomycin and 100 U/mL penicillin (Sangon Biotech, Shanghai, China) at 37 °C and 5% CO_2_. RNA was extracted according to the previous experimental method and detected by RT-PCR [27]. The primer sequences were as follows:

GAPDH, F: 5′-CAGGAGGCATTGCTGATGAT-3′,

R: 5′-GAAGGCTGGGGCTCATTT-3′.

LINC00460, F: TCGGCTAAGAGTCACCCTGGATG-3′, 

R: 5′- CACAGACGCCTCCCACACAATG-3′.

MCM4, F: 5′- ATCTCCCTCTCAGAGACGTAG-3′ 

R: 5′-TGTCAGTGGTGAACTAACATCA-3′.

U6, F: 5′-AGAGAAGATTAGCATGGCCCCTG-3′,

R: 5′-AGTGCAGGGTCCGAGGTATT-3′.

miR-338-3p, F: 5′-CGCGTCCAGCATCAGTGATT-3′,

R: 5′-AGTGCAGGGTCCGAGGTATT-3′.

### 2.6. Hematoxylin-Eosin (HE) Staining and Immunohistochemistry 

LUAD tumor tissue and paracancerous tissue were obtained from the Department of Hematology & Oncology, the First Hospital of Changsha. This study was approved by the ethics committee. The collection of tissue specimens was explained in detail to the patients or their families in advance, and an “informed consent form” was signed after obtaining the consent of the patients and their families. Tissues were fixed and embedded in paraffin, then cut into 5 μm thick sections and mounted on New Silane slides with a MCM4 mouse polyclonal antibody (Cat.# D260599, Sangon Biotech, Shanghai, China). Standard hematoxylin-eosin (HE) staining and immunohistochemical (IHC) were used to assess protein expression levels in tumor samples [28].

### 2.7. Enrichment Analysis and Interaction Network

The top 200 genes associated with the MCM4 gene in LUAD were gained from GEPIA (http://gepia.cancer-pku.cn/, accessed on 28 March 2022). The Gene Ontology (GO) and Kyoto Encyclopedia of Genes and Genomes (KEGG) were analyzed by DAVID (https://david.ncifcrf.gov/tools.jsp, accessed on 28 March 2022) and visualized by “ggplot2”. The functional association network of the target gene was predicted by GeneMINIA (http://genemania.org/, accessed on 28 March 2022). The “Multiple proteins” module of the STRING database (https://string-db.org/, accessed on 29 March 2022) was used to map the top 10 protein interaction networks associated with the target genes based on the same function that the proteins co-promote.

### 2.8. Mutation Analysis

Through cBioPortal (https://www.cbioportal.org/, accessed on 5 April 2022), the mutation rate, types and common mutation sites of MCM4 in tumors were acquired, and the relationship between MCM4 mutation and clinical prognosis of patients was explored. 

### 2.9. Targeted Drug Analysis of MCM4

We hope to address the problem of EGFR target resistance through the treatment of downstream genes of the EGFR. We performed a drug target analysis of the MCM4 gene through the HERB (http://herb.ac.cn/, accessed on 5 April 2022) and GSCAlite (http://bioinfo.life.hust.edu.cn/web/GSCALite/, accessed on 5 April 2022) databases. The expectation is that these targeted drugs may play a role in the treatment of LUAD. 

### 2.10. Immune Correlates

Correlations between gene expression levels and tumor mutational burden (TMB) were evaluated based on the mutation data of LUAD downloaded from TCGA. We also used “Estimate” in the R package to score the stromal and immune cells of the samples, where the stromal score represents the number of stromal cells in the tumor immune microenvironment (TIME) and the immune score represents the number of immune cells in TIME; the sum of these two is the total score.

### 2.11. Statistical Analysis

All statistical analyses were performed by GraphPad Prism 8.4.3 and SPSS version 23.0. Statistically significant differences between the two groups of data were estimated by the Mann–Whitney test and independent t-test. One-way ANOVA was used to assess statistical differences between multiple groups of data. A *p*-value < 0.05 was considered statistically significant.

## 3. Results

### 3.1. Expression and Prognostic Value of EGFR in LUAD

EGFR overexpression in lung cancer tissue was found based on the Human Protein Atlas database (HPA, http://www.proteinatlas.org/, accessed on 15 January 2021) (Figure 1A and Appendix A). In addition, the cBioPortal database (http://www.cbioportal.org/, accessed on 15 January 2021) showed that changes in EGFR gene expression in the TCGA LUAD dataset were mainly due to its amplification and missense mutations (Figure 1B). The increase in gene copy numbers was likely to be one of the main mechanisms that made a contribution to the over-regulation of the EGFR in LUAD patients (Appendix A). Similar EGFR expression imbalances were confirmed by IHC from the HPA database (Figure 1C), and patients’ information was shown in Appendix A. The EGFR was highly and differentially expressed in tumor samples between tumor-paracancer (*p* = 0.0301) and paired tumor-normal (*p* = 0.0451) samples (Figure 1D,E). In addition, some traditional prognostic factors were also analyzed (Figure 1F–J), and those results displayed a significant correlation between tumor metastasis and high expression of the EGFR (*p* = 0.0057) (Figure 1I). The high EGFR expression could be closely related to tumor metastasis in LUAD. 

### 3.2. Identification of Differentially Expressed Genes

Based on the above analysis, the ceRNA network associated with the EGFR can be used as a potential prognostic model for patients of LUAD. Moreover, we must make it clear that the meaning of the expression levels in LUAD samples with EGFR^high^ and EGFR^low^ expression groups are consistent with those in cancer and paracancerous groups. In total, 1098 DElncRNAs, 36 DEmiRNAs and 4738 DEmRNAs were screened in LUAD samples with EGFR^high^ and EGFR^low^ expression groups. The distribution of DElncRNAs, DEmiRNAs and DEmRNAs were visualized by the volcano plot as shown in Figure 2A–C.

### 3.3. ceRNA Regulatory Network

A total of 62 lncRNAs and their potential target 4 miRNAs were identified based on the TarBase database. Meanwhile, the target mRNAs of these 4 miRNAs were identified by TargetScan and miRDB databases. Ultimately, 376 of the 4738 DEmRNAs were identified. Finally, Cytoscape software was used to construct and visualize the EGFR-related lncRNA-miRNA-mRNA triple regulatory network in LUAD, including 62 lncRNAs, 4 miRNAs and 376 mRNAs (Figure 2D). In order to further explore the potential functions of the ceRNA network related to the EGFR, a functional enrichment analysis (including GO and KEGG) was performed on these mRNAs by Metascape (Figure 2E).

In order to determine whether these RNAs were related to prognosis in LUAD, we first performed an OS analysis of LUAD patients using a Kaplan–Meier analysis and log-rank test. In total, 7 DElncRNAs, 2 DEmiRNAs and 37 DEmRNAs were found to be associated with prognosis (Figure 3). Comparing the lncRNA-miRNA and miRNA-mRNA target gene matching results from the above analysis, a survival-related ceRNA regulatory network including seven lncRNAs, two miRNAs and eight mRNAs were finally constituted (Figure 2F). 

### 3.4. Construction of a ceRNA Network Model with LUAD Prognostic Specificity 

To identify ceRNA networks of significant prognostic value in LUAD, we further explored RNA expression levels in high and low EGFR expression groups as well as in tumor and adjacent normal lung tissue (Figure 4A). The results showed six upregulated (AC084083.1, ARHGEF26-AS1, VIPR1-AS1, LINC00342, MEG3, LINC00460) and one downregulated (AP002478.1) lncRNAs, two downregulated (mir-215, mir-338-3p) miRNAs, seven upregulated (MCM4, CDIP1, RAB27B, TMEM255A, LSAMP, ATP8B4, SRGAP3) and one undifferentiated (CLCN3) mRNAs in LUAD samples with EGFR^high^ and EGFR^low^ groups (Figure 4A). In addition, the expression levels of these RNAs were confirmed in 54 (or 46) paired LUAD samples (Figure 4B), as well as in 59 (or 46) normal samples and 535 (or 521) LUAD samples (Appendix A).

Then, the LINC00460 (upregulated)—mir-338-3p (downregulated)-MCM4 (upregulated) axis was finalized according to the results of the expression validation of DERNAs and survival analysis. It was found that the expression of LINC00460 was negatively correlated with the expression of mir-338-3p and positively correlated with the expression of MCM4 by expression correlation analysis (Figure 5A). Pairing between mir-338-3p and the target sites in LINC00460 and MCM4 was predicted by MiRcode and TargetScan, respectively (Figure 5B). These data indicate that LINC00460 can enhance the expression of MCM4 by sponge adsorption mir-338-3p. 

### 3.5. Effect of Different Expression Patterns of LINC00460-mir-338-3p-MCM4 Axis on OS of LUAD

We have creatively developed a new analytical model to explore the correlation between different combinations of expression patterns of three genes (LINC00460, mir-338-3p and MCM4) with LUAD patients and OS. The results indicated that the OS of patients in the pattern of L^−^/m^+^/M^-^ was markedly improved compared to the expression pattern of L^+^/m^−^/M^+^ (*p* = 0.0093) (Figure 5C,D). Furthermore, other expression modes showed that the OS of LUAD patients in the pattern of L^−^/m^+^/M^−^ was markedly improved compared with the pattern of L^+^/m^+^/M^+^ and L^+^/m^−^/M^−^. The results manifested that the downregulation of LINC00460 and MCM4 expression levels, or the promotion of mir-338-3p expression, could inhibit the occurrence and development of LUAD cells. These were consistent with the above analysis results, illustrating the feasibility and accuracy of our new analysis model.

### 3.6. Clinical Relevance of LINC00460/MCM4 Axis in LUAD Patients

To understand whether the expression levels of LINC00460 and MCM4 were influenced by clinical characteristics, the correlation was explored between LINC00460 and MCM4 expression with clinical factors. These results indicated that the expression of both LINC00460 and MCM4 were positively correlated with lymph node metastasis (LINC00460, *p* = 0.0006; MCM4, *p* = 0.0463) and TNM stage (LINC00460, *p* = 0.0468; MCM4, *p* = 0.0356) (Table 1), and MCM4 expression was also positively correlated with distant metastasis (*p* = 0.021). In addition, MCM4 expression values were significantly higher in men than in women (*p* = 0.0006) and higher in patients aged <60 than in patients aged ≥60 (*p* = 0.012, Table 1). We also detected the correlation of LINC00460 and MCM4 with the overall survival and TNM stage of LUAD patients in the GEPIA database. The results revealed that the high expression of LINC00460 and MCM4 was significantly related to the poor prognosis of LUAD patients (*p* < 0.01, Appendix A), which is consistent with our analysis results.

Furthermore, univariate and multivariate Cox regression analyses were used to find out the correlation between clinical features and OS. In the univariate Cox regression analysis model of LINC00460 and MCM4, clinical prognostic factors such as TNM stage, tumor size and lymph node metastasis of LUAD patients were closely related to OS (*p* < 0.05; Table 2 and Table 3). Importantly, both LINC00460 (HR = 1.383, *p* = 0.043) and MCM4 (HR = 1.608, *p* = 0.003) over-expression significantly related to a worse prognosis (Table 2 and Table 3). The multivariate Cox regression analysis of MCM4 indicated that tumor size (HR = 1.550, *p* = 0.012), lymph node metastasis (HR = 1.974, *p* < 0.0001) and high expression of MCM4 (HR = 1.459, *p* = 0.019) were closely related to OS in LUAD patients (Table 3). In summary, MCM4 may serve as an important prognostic factor for patients with LUAD.

### 3.7. Validation of LINC00460/MCM4 Axis Expression Levels

The expression levels of the LINC00460-miR-338-MCM4 regulatory axis in lung adenocarcinoma cells (A549, PC-9 and H1299) and normal human bronchial epithelial cells (BEAS-2B) were detected by RT-PCR (Figure 6). The expression levels of LINC00460 and MCM4 were significantly up-regulated in A549, PC-9 and H1299 relative to BEAS-2B cells, while the expression of miR-338 was significantly down-regulated (*p* < 0.05). This confirmed the accuracy of the bioinformatics analysis results. In addition, the expression of the MCM4 protein in lung adenocarcinoma tumors and paracancerous tissue was also examined by IHC (Figure 6D), and the results indicated that MCM4 was significantly highly expressed in LUAD tumor tissues.

### 3.8. Enrichment Analysis and Interaction Network of MCM4-Related Genes

In order to further explore the possible function of MCM4 in LUAD, GO and KEGG enrichment analyses were performed on the top 200 MCM4-related genes in LUAD (Figure 7). The enrichment terms associated with MCM4 were “p53 signaling pathway” and “cell cycle”. In addition, a GO enrichment analysis indicated that MCM4 mainly enriched in “cell division”, “mitotic nuclear division”, “nucleoplasm” and “protein binding”.

The functional association network of MCM4 and EGFR was predicted by the GeneMINIA tool and 20 potential target genes were shown (Figure 7E,F). MCM2, MCM3, MCM5, MCM6, MCM7 and MCM4 belong to the MCM family and are closely related to the EGFR. MCM proteins are essential proteins for initiation and elongation steps during DNA replication in organisms. This suggests that MCM4 may affect the occurrence and development of LUAD by regulating processes such as DNA replication and the cell cycle.

### 3.9. Potential Drugs Targeting MCM4

An HERB database analysis identified 4 drugs with MCM4 as a potential target (Figure 8A). Furthermore, we used the GSCALite database and assessed the correlation of drug sensitivity with MCM4 according to the Cancer Therapy Response Portal (CTRP) and genomics of drug sensitivity in cancer (GDSC). The CTRP illustrated that MCM4 was resistant to Docetaxel and sensitive to 17 drugs including 5-Fluorouracil (5-fluorouracil), PHA-793887 and KIN001-102 (Figure 8B). Figure 8C shows that the high expression level of MCM4 was also sensitive to the other 138 drugs. These drugs may target MCM4 and play a role in the treatment of LUAD. 

In addition, the cBioPortal tool was used to analyze the mutational signature of MCM4. The OncoPrint plot showed the amplification of MCM4 genomic in the TCGA LUAD dataset in Appendix A. A significant association was observed between MCM4 expression and copy number in LUAD samples (Appendix A). Therefore, these data indicated that the aberrant expression of MCM4 was mainly due to copy number changes in LUAD rather than gene mutations. Appendix A illustrated several highly mutated sites in the MCM4 gene and their positions in the three-dimensional structure. In addition, it can also be seen that the most mutated forms of MCM4 in pan-cancer cells are also amplified and missense mutations (Appendix A). This suggested that the pro-oncogenic mechanism of MCM4 in LUAD was similar to that of other pan-cancer mechanisms.

### 3.10. Correlation between Immune Infiltration and Expression of MCM4 in LUAD

An iammune correlation analysis displayed that the expression levels of six immune markers of neutrophils (CD66b) and dendritic cells (HLA-DPB1, HLA-DQB1, HLA-DRA, HLA-DPA1 and BDCA-1) have significant negative correlations with MCM4 expression in LUAD. In addition, MCM4 expressions have significant negative correlations with the expression levels of four immune markers of three immune cells, including the natural killer cell (KIR2DL4), Th1 (STAT1, IFN-γ) and T cell exhaustion (GZMB) (Appendix A). We also verified the relationship between MCM4 expression levels in LUAD and the above-mentioned markers using the GEPIA database and obtained similar results (Appendix A).

Tumor mutation burden (TMB) represents the number of mutations in tumor cell genes per million genes in a patient sample. The correlation results between TMB and MCM4 showed that the expression level of MCM4 correlated with TMB (Appendix A). TIME values for LUAD samples were assessed and compared with gene expression and patient OS (Appendix A). The results showed that there was a significant negative correlation between MCM4 expression and stromal score, immune score and total score. The lower the content of stromal cells and immune cells in the tumor environment, the higher the purity of the tumor. These results demonstrated that the LINC00460/MCM4 axis may regulate the level of tumor infiltrating immune cells in LUAD, which in turn has an impact on clinical outcomes.

## 4. Discussion

The EGFR gene regulates many physiological processes and induces important mechanisms associated with cancer [29,30]. Previous studies have found that a high expression and high copy number of the EGFR were demonstrated in 10–30% of NSCLC patients [31]. Earlier studies suggested that constitutive activation of the EGFR signaling pathway in tumor tissues may be initiated by EGFR gene amplification or triggered by EGFR mutations [5,32,33]. In contrast, Li et al. found that EGFR over-expression was mainly closely related to amplification but statistically independent of EGFR mutations in lung adenocarcinoma [34]. The specific mechanism of EGFR action in tumors has not been clarified and further studies are needed. In this study, we obtained the ceRNA regulatory network associated with EGFR expression by computer analysis. Comprehensive bioinformatics analysis strongly suggests that the LINC00460-mir-338-3p-MCM4 ceRNA network plays an important role in the diagnosis and prognosis of LUAD. 

The analysis results of LINC00460, mir-338-3p and MCM4 were basically consistent with the existing research reports [35,36]. LncRNAs may serve as potential predictive and prognostic markers for EGFR resistance in LUAD, as they are involved in modulating chemosensitivity, radiosensitivity and sensitivity to EGFR-targeted therapy through multiple mechanisms [37]. Studies have revealed that LINC00460 is significantly upregulated in NSCLC and promotes the metastasis and invasion of lung cancer cells by inducing epithelial–mesenchymal transformation [35]. The expression of LINC00460 is up-regulated in gefitinib-resistant NSCLC tissues and cells and is closely associated with advanced tumor stage and poor clinical prognosis. More interestingly, LINC00460 promoted EGFR expression by sponging miR-769-5p, thereby promoting the resistance of NSCLC cells to gefitinib [38]. LINC00460 may be a novel prognostic and therapeutic target for LUAD. miR-338-3p was poorly expressed in NSCLC tissue relative to adjacent noncancerous tissue. LUAD cell proliferation, migration and invasion were inhibited by miR-338-3p overexpression. miR-338-3p directly targets Neuropilin 1 (NRP1) and plays a role in enhancing drug sensitivity in EGFR wild-type NSCLC patients [39]. The mini chromosome maintenance (MCM) protein 2–7 forms the complex necessary for DNA replication to begin [40]. Kikuchi et al. [36] found that the expression level of MCM4 in NSCLC cancer cells was higher than in adjacent normal lung cells (*p* < 0.001), and MCM4 might play a significant role in the proliferation of NSCLC cells. However, the mechanism of action between MCM4 and EGFR has not been clearly reported.

In this research, the univariate Cox regression analysis of LINC00460 and MCM4 indicated that TNM stage, tumor size and lymph node metastasis were closely related to the OS of LUAD patients. Importantly, both LINC00460 and MCM4 over-expression significantly related to a worse prognosis. A multivariate Cox regression analysis of MCM4 showed that tumor size, lymph node metastasis and MCM4 high expression were also associated with OS in LUAD patients. A multi-gene regulation model analysis revealed that the L^−^/m^+^/M^−^ pattern significantly improved the OS compared to the L^+^/m^−^/M^+^ expression pattern in LUAD patients (*p* = 0.0093). This suggests that inhibiting the expression of LINC00460 and MCM4 or upregulating the expression of mir-338-3p can both prevent tumor progression and improve the prognosis of LUAD patients.

In addition, the approach of this study provides a new idea to address tumor drug resistance. The research has indicated that RAS dysregulation and the resulting signaling dysregulation account for one-third of all human cancers, and mutations in RAS are usually related to treatment resistance and poor prognosis [41,42]. As a key cancer driver, RAS has always been the focus of an intensive search for therapeutic approaches. So far, however, no effective RAS inhibitors have been approved for clinical use. Recently, the clinical results of KRAS G12C inhibitors have sparked excitement in the scientific community [43,44]. Nevertheless, acquired drug resistance may limit the efficacy of inhibitors, indicating that combination therapy may be required [45]. By analyzing RAS-related dysregulated genes and constructing a RAS-centered ceRNA network, oncogenic RAS and its downstream signaling and metabolic programs can be more effectively targeted. An accurate understanding of the coordinated interactions between RAS and other genes in the associated ceRNA network will be very important for developing novel targeted therapies for RAS-driven cancers.

In conclusion, we have determined that the ceRNA-based LINC00460/MCM4 axis may be a potential therapeutic target and prognostic biomarker of LUAD, but there are still some limitations. First, most of our study data were obtained from the TCGA database, and some of the analysis results may be biased and need to be validated by further clinical trials. Secondly, the related mechanisms of the LINC00460/MCM4 axis and EGFR gene in LUAD need to be further investigated experimentally. Finally, the specific mechanism of action between the LINC00460/MCM4 pathway and EGFR requires further experimental studies. Despite these deficiencies, a LINC00460—mir-338-3p—MCM4 regulatory network was identified by comprehensive analysis of the EGFR and ceRNA, which is expected to be an effective diagnostic and therapeutic target.

## 5. Conclusions

A detailed comprehensive analysis of the EGFR and ceRNA was performed, indicating that LINC00460/MCM4 can effectively predict the survival outcome of patients with LUAD and can hopefully be an effective procedure for diagnosis and treatment. All in all, this research further investigates the molecular pathogenesis of LUAD, which can be served to instruct in-depth research of LUAD. 

## Figures and Tables

**Figure 1 cancers-14-05073-f001:**
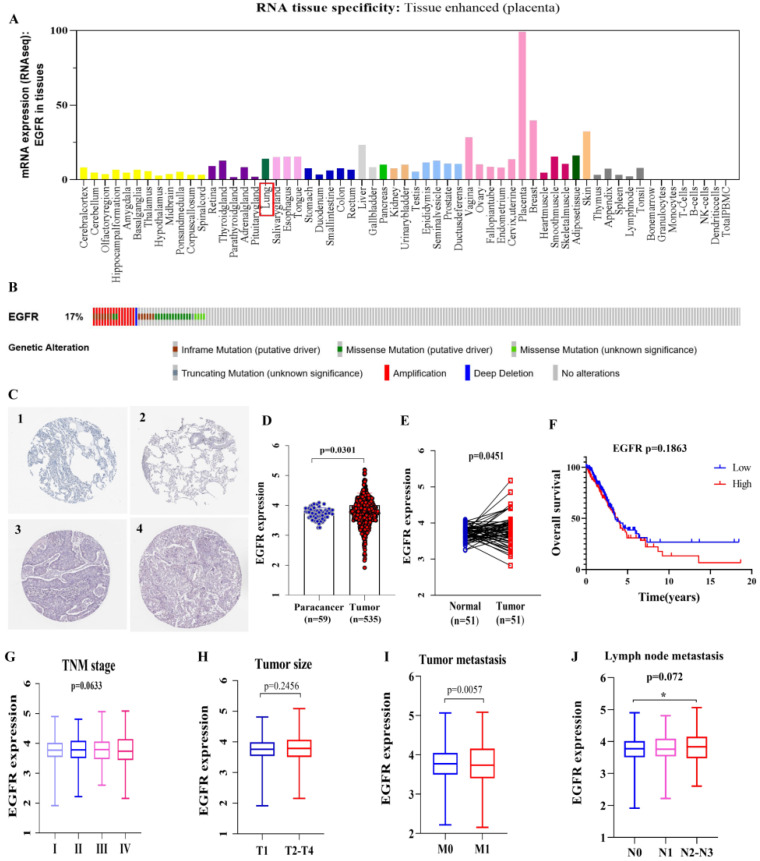
High expression of EGFR in LUAD and prognostic value. (**A**) EGFR expression distribution in whole cancer tissues. (**B**) cBioPortal OncoPrint plot displaying the distribution of EGFR genomic alterations rate in LUAD dataset. (**C**) Validation of EGFR expression at the translational level by HPA database, normal tissue (1 & 2), and tumor tissue (3 & 4). (**D**,**E**) EGFR expression in tumor- paracancer paired and tumor-normal samples. (**F**) The low expression (*n* = 243) and high expression (*n* = 244) of EGFR were compared by the Kaplan–Meier survival curve. (**G**) Correlation of EGFR expression with TNM stage, stage I, *n* = 272; stage II, *n* = 119; stage III, *n* = 84; stage IV, *n* = 25. (**H**) Correlation between EGFR expression and tumor size, T1, *n* = 166; T2-T4, *n* = 337. (**I**) Correlation between EGFR expression and tumor lymph node metastasis, M0, *n* = 339; M1, *n* = 23. (**J**) Correlation between EGFR expression and distant metastases, N0, *n* = 325; N1, *n* = 93; N2-N3, *n* = 76. The RNAs expression value was logarithmized with log10. Statistically significant differences between the two groups of data were estimated by the Mann–Whitney test and independent t-test. One-way ANOVA was used to assess statistical differences between multiple groups of data. A * *p* < 0.05. *p* < 0.05 was considered statistically significant.

**Figure 2 cancers-14-05073-f002:**
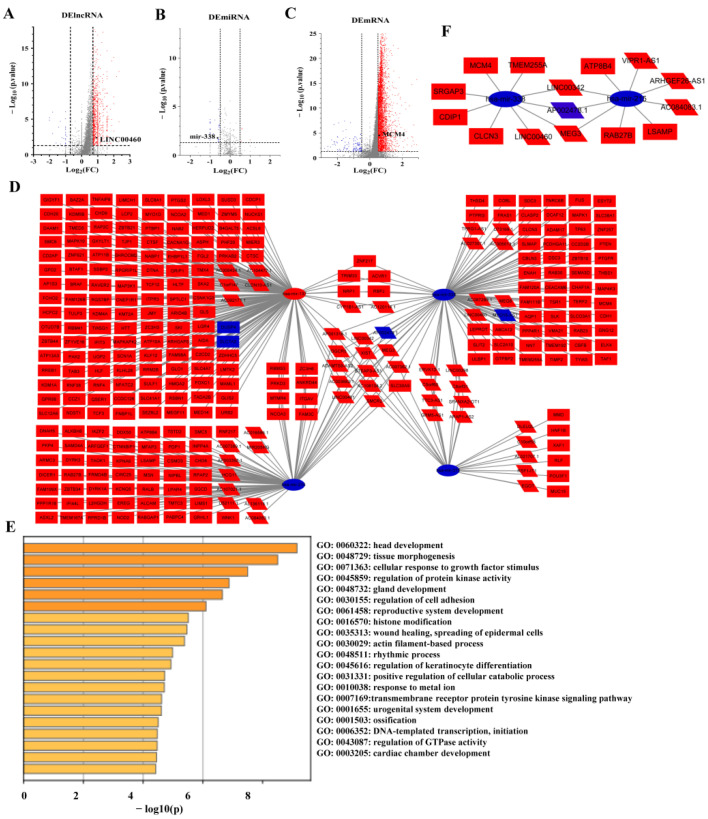
Volcano map of DERNAs between EGFR^high^ and EGFR^low^ expression in LUAD samples and construction of ceRNA network. Red indicates upregulation, blue represents downregulation. (**A**) Volcano plots of differentially expressed lncRNAs (DElncRNA) (|log_2_FC| > 0.70, *p* < 0.05). (**B**) Volcano plots of differentially expressed miRNAs (DEmiRNA) (|log_2_FC| > 0.50, *p* < 0.05). (**C**) Volcano plots of differentially expressed mRNAs (DEmRNA) (|log_2_FC| > 0.50, *p* < 0.05). (**D**) The triple regulatory network in LUAD. (**E**) Functional enrichment analysis of mRNAs in the network. (F) Survival-related DERNAs regulatory network in LUAD.

**Figure 3 cancers-14-05073-f003:**
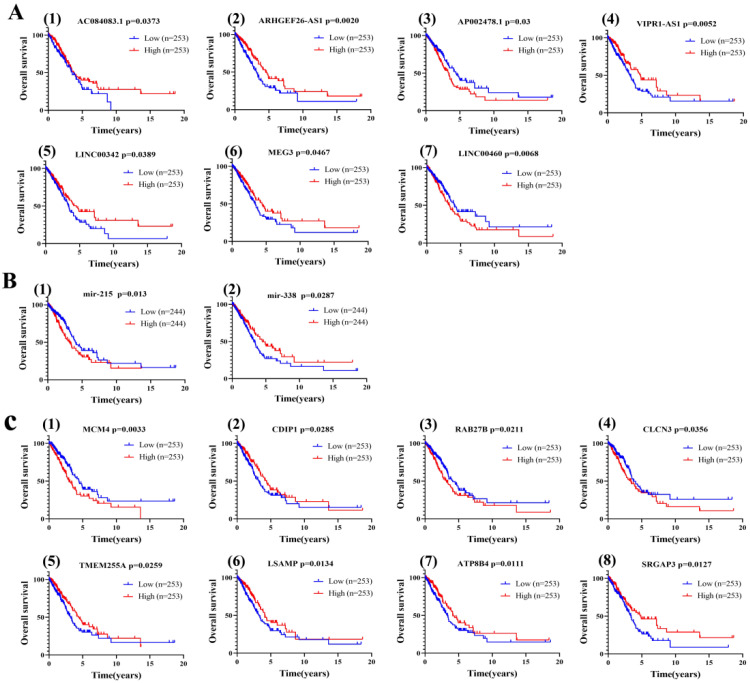
Correlation analysis of differentially expressed RNAs in ceRNA network with overall survival of LUAD patients by a Kaplan–Meier survival curve. (**A**) lncRNAs. (**B**) miRNAs. (**C**) mRNAs. Statistically significant differences between the two groups of data were estimated by the Mann–Whitney test. A *p* < 0.05 was considered statistically significant.

**Figure 4 cancers-14-05073-f004:**
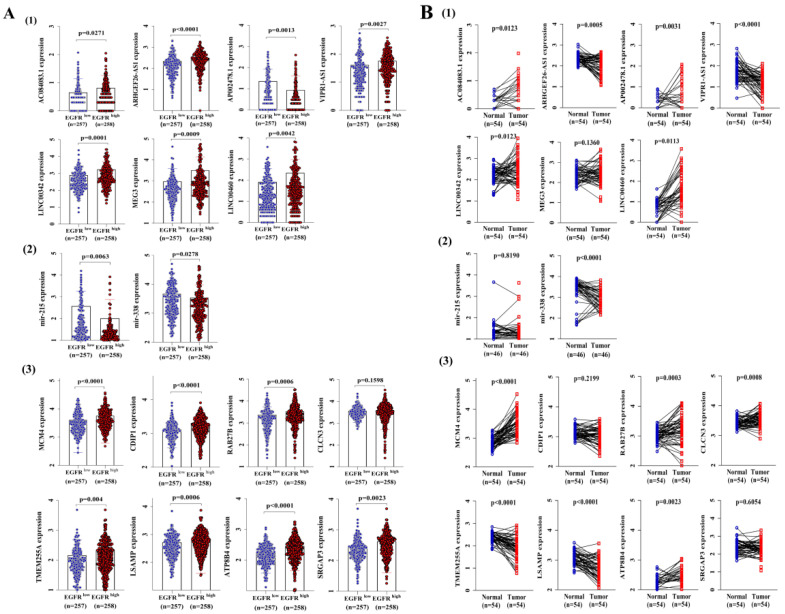
The distribution of 17 RNAs’ expression values from the ceRNA network in the TCGA LUAD dataset. (**A**) The expression of 17 RNAs in LUAD samples with EGFR^high^ and EGFR^low^ expression groups. (**B**) The distribution of 17 RNAs’ expressions in paired LUAD tissues. The RNAs’ expression values were logarithmized with log10. Statistically significant differences between the two groups of data were estimated by an independent *t*-test. A *p* < 0.05 was considered statistically significant.

**Figure 5 cancers-14-05073-f005:**
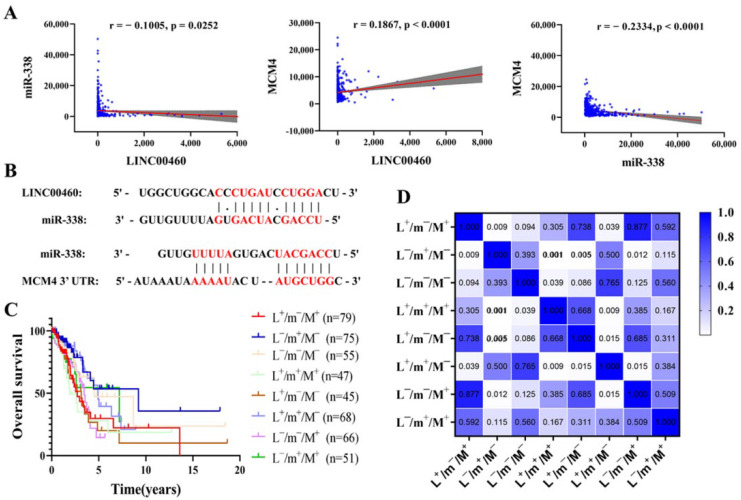
Target site prediction and expression correlation analysis among LINC00460, mir-338-3p and MCM4. (**A**) Correlation analysis between LINC00460, mir-338-3p and MCM4 in LUAD (*n* = 515). Grey shaded areas indicate 95% confidence intervals. (**B**) Base pairing between mir-338-3p and the target site in the LINC00460 and MCM4 3′ UTR. Correlation analysis of different expression patterns of LINC00460—mir-338-3p—MCM4 with OS of LUAD patients (**C**), with heat map of significance of differences (**D**). A *p* < 0.05 was considered statistically significant.

**Figure 6 cancers-14-05073-f006:**
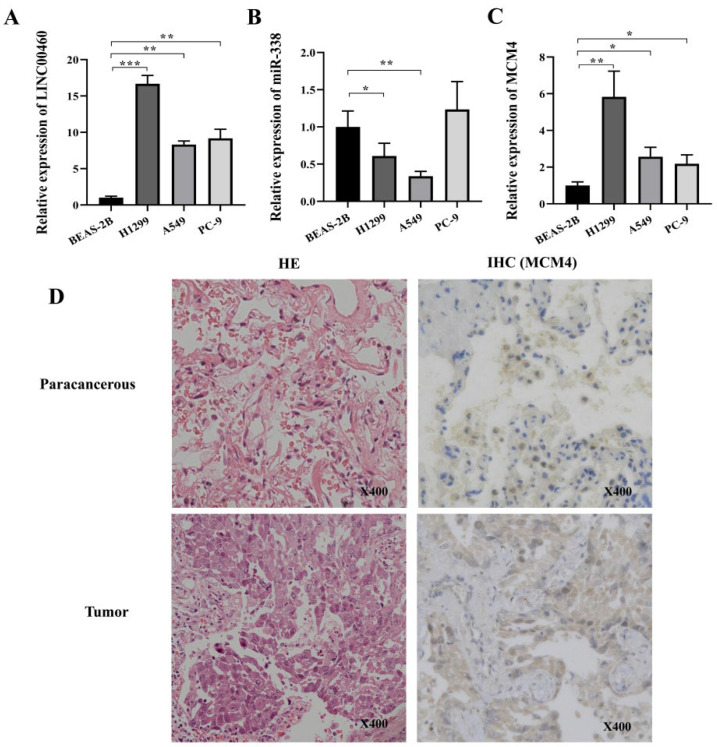
Differential gene expression levels were verified by RT-PCR and IHC (MCM4). The expression level of LINC00460 (**A**), miR-338 (**B**), MCM4 (**C**) in BEAS-2B, A549, PC-9 and H1299 cells was detected by RT-PCR. (**D**) Tumor and paracancerous tissue sections were subjected to H&E staining and IHC against MCM4. Statistically significant differences between the two groups of data were estimated by independent t-test. * *p* < 0.05, ** *p* < 0.01, *** *p* < 0.001. *p* < 0.05 was considered statistically significant.

**Figure 7 cancers-14-05073-f007:**
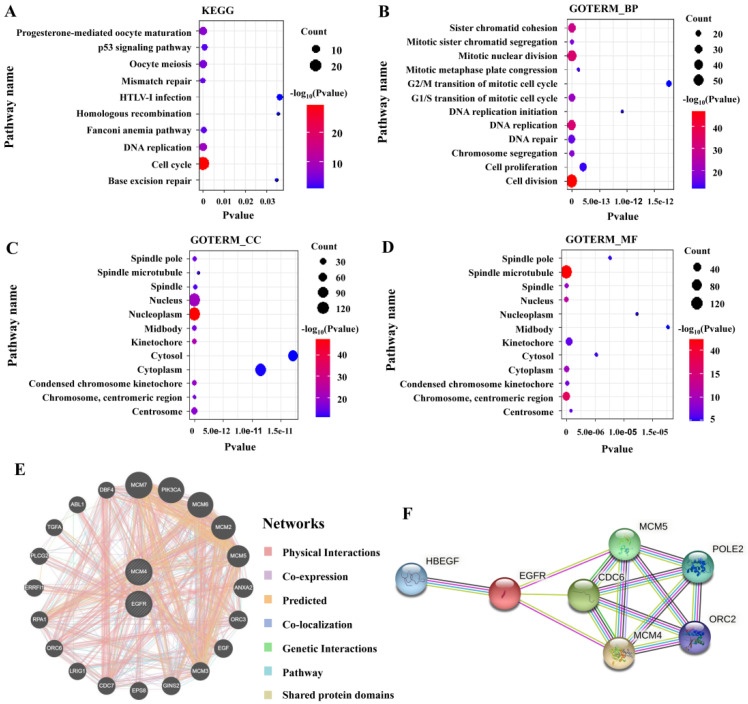
Functional enrichment analysis of MCM4-related genes in LUAD as well as the interaction network between MCM4 and EGFR. (**A**) KEGG, (**B**) biological process (BP), (**C**) cellular component (CC), (**D**) molecular function (MF), (**E**) the gene interaction network between MCM4 and EGFR, (**F**) the protein interaction network between MCM4 and EGFR.

**Figure 8 cancers-14-05073-f008:**
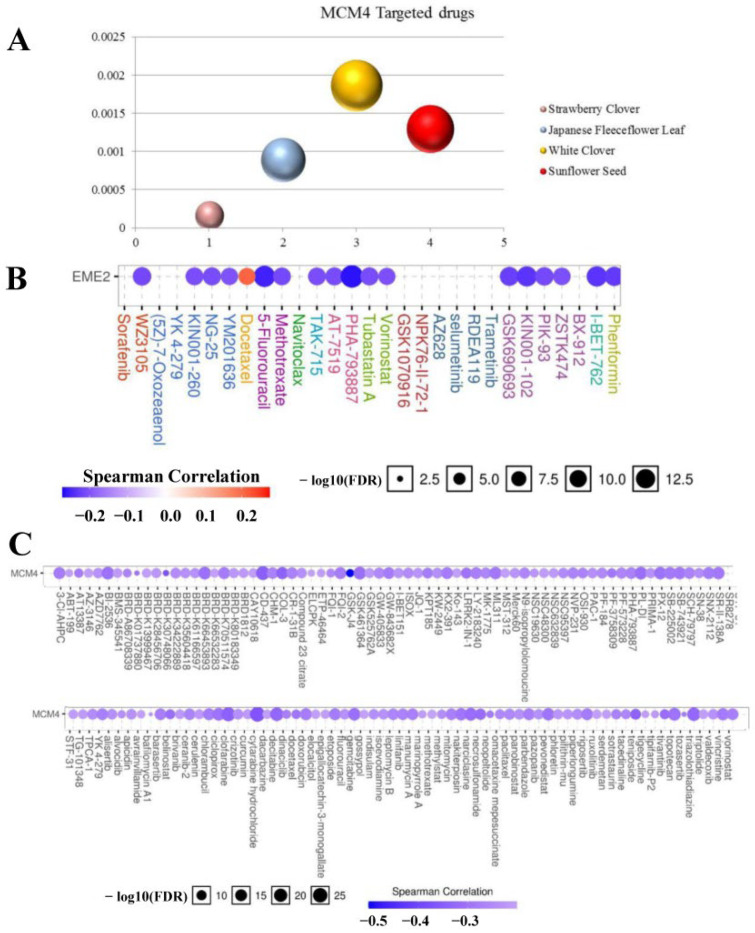
Possible targeted drugs of MCM4. (**A**) Acquiring the targeted drug of MCM4 through HERB database. (**B**) Evaluate the correlation between drug sensitivity and MCM4 by CTRP. (**C**) Evaluate the correlation between drug sensitivity and MCM4 by GDSC.

**Table 1 cancers-14-05073-t001:** Univariate analysis of overall survival in LUAD patients stratified based on clinical characteristics.

Factor	Variable	N	LINC00460 Expression (Median)	*p*-value	MCM4 Expression (Median)	*p*-Value	Overall Survival		
							Months (Mean)	95% CI (Mean)	*p*-Value (Log-Rank Test)
Age	≥60	353	20	0.889	3172	**0.012**	24.647	22.053–27.241	0.143
	<60	134	20		3690		27.515	21.747–33.283	
Gender	Male	226	17	0.112	3921.5	**0.0006**	25.814	21.824–29.804	0.984
	Female	261	22		3077		25.109	22.086–28.132	
Tumor size (T)	>4	91	27	0.245	3455	0.1273	18.420	15.265–21.575	**0.0007**
	≤4	393	19		3383		27.110	24.181–30.04	
Lymph node metastasis (N)	Negative	313	16	**0.0006**	3282	**0.0463**	27.008	23.714–30.303	**<0.0001**
	Positive	162	35		3638		22.988	19.299–26.676	
	Unknown	12	20		2112.5		17.478	10.683–24.273	
Distant metastasis (M)	Negative	320	23	0.119	3547.5	**0.021**	25.963	22.97–28.956	**0.027**
	Positive	23	24		6456		23.494	15.296–31.692	
	Unknown	144	15		3157.5		24.576	19.724–29.428	
TNM stage	I–II	376	18	**0.0468**	3306.5	**0.0356**	26.752	23.756–29.747	**<0.0001**
	III–IV	104	25		3755		19.856	16.506–23.206	
	Unknown	7	57		3306		37.667	9.879–66.355	

*p* < 0.05 were marked in bold to highlight differences.

**Table 2 cancers-14-05073-t002:** Univariate and Multivariate analyses (Cox regression model) of LINC00460 in LUAD patients.

Factor	Univariate Cox			Multivariate Cox		
	HR	95% CI	*p*-Value (Log-Rank Test)	HR	95% CI	*p*-Value (Log-Rank Test)
Age	1.078	0.761–1.528	0.671			
Gender	0.99	0.725–1.351	0.948			
TNM stage	1.813	1.398–2.350	**<0.0001**			
Tumor size(T)	1.958	1.390–2.757	**<0.0001**	1.543	1.101–2.163	**0.012**
Lymph node metastasis	2.219	1.704–2.890	**<0.0001**	2.051	1.563–2.690	**<0.0001**
Distant metastasis	0.954	0.798–1.139	0.602			
LINC00460 expression (high/low)	1.383	1.011–1.893	**0.043**			

*p* < 0.05 were marked in bold to highlight differences.

**Table 3 cancers-14-05073-t003:** Univariate and Multivariate analyses (Cox regression model) of MCM4 in LUAD patients.

Factor	Univariate Cox			Multivariate Cox		
	HR	95% CI	*p*-Value (Log-Rank Test)	HR	95% CI	*p*-Value (Log-Rank Test)
Age	1.078	0.761–1.528	0.671			
Gender	0.99	0.725–1.351	0.948			
TNM stage	1.813	1.398–2.350	**<0.0001**			
Tumor size(T)	1.958	1.390–2.757	**<0.0001**	1.550	1.103–2.177	**0.012**
Lymph node metastasis	2.219	1.704–2.890	**<0.0001**	1.974	1.498–2.600	**<0.0001**
Distant metastasis	0.954	0.798–1.139	0.602			
MCM4 expression (high/low)	1.608	1.176–2.200	**0.003**	1.459	1.064–2.000	**0.019**

*p* < 0.05 were marked in bold to highlight differences.

## Data Availability

The data involved in this study were included in this manuscript and Appendix A.

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
