# Peer review of "Identification of EGFR-Related LINC00460/mir-338-3p/MCM4 Regulatory Axis as Diagnostic and Prognostic Biomarker of Lung Adenocarcinoma Based on Comprehensive Bioinformatics Analysis and Experimental Validation"

_cancers, 2022, doi:10.3390/cancers14205073_

Round 1

Reviewer 1 Report

An impressive article by Dr. Li and Dr. Bai identifies EGFR‐related LINC00460/mir-338-3p/MCM4 regulatory axis as a diagnostic tool by exploring bioinformatic analysis. This is a very well-written manuscript following the hypothesis of the work. Though few things need to be addressed before it is ready for acceptance. They are as follows:

1.  Some of the figure labelings are not at all readable. please work with them before resubmission. See Fig 2 and 4 including others for details.

2. Authors should add a few lines in the introduction explaining how EGFR mutation mediates resistance to EGFR tyrosine kinase inhibitors (PMID: 33775864). This will give an idea to readers of how significant is the finding from this manuscript in the case of the translational aspect.

3. It has been shown recently that 20% of Lung cancer has RAS mutation (PMID: 33870211) while 22% of LAUD has KRAS mutation. Authors should discuss one of the future aspects of this study is RAS-mediated pathways in LAUD which might be involved in creating resistance. The discovery of KRAS G12C inhibitor resistance is one of the fields in lung cancer that is more concerned. The approach taken by this manuscript has some direct future implications in RAS therapy. The authors should add a few lines discussing this aspect.  

Author Response

Dear professor:

I would like to thank you for your helpful review of my paper. Your comments are extremely useful, and I really appreciate your kindness. We have studied the comments carefully and made corrections which we hope meet with approval. The manuscript has modified according the comments.

Sincerely

Wen Li

2022.10.6

Responses to Reviewer

Q1) Some of the figure labelings are not at all readable. please work with them before resubmission. See Fig 2 and 4 including others for details.

A1) Thank you for your comment. We've rescaled the all image labels to make them clearer.

Q2) Authors should add a few lines in the introduction explaining how EGFR mutation mediates resistance to EGFR tyrosine kinase inhibitors (PMID: 33775864). This will give an idea to readers of how significant is the finding from this manuscript in the case of the translational aspect. 

A2) Thank you for your comment. We have added a description of how EGFR mutations can mediate resistance to EGFR tyrosine kinase inhibitors (EGFR-TKIs) to the 'introduction' section. “And most EGFR mutations in NSCLC occur in the exons of the receptor tyrosine kinase domain. Although EGFR tyrosine kinase inhibitors (EGFR-TKIs), including Erlotinib and Gefitinib, have shown initial efficacy in 30% of NSCLC patients with EGFR mutations in the past few decades, secondary resistance often occurs in EGFR-TKIs treatment. Currently, Multiple mechanisms of secondary resistance to EGFR-TKIs, including primary or secondary T790M point mutations, human epidermal growth factor receptor 2 (HER2) amplification, mesenchymal epithelial cell transforming factor (MET) amplification or activation of bypass signaling pathways by phosphatidylinositol 3 kinase (PI3K) mutations and epithelial-mesenchymal transitions (EMT) have been clarified. However, the combination of EGFR-TKIs with platinum or other cytotoxic chemotherapeutic agents did not achieve the expected prolongation of survival in NSCLC patients, resulting in increased toxic and side effects. Therefore, it is of great significance to deeply study the molecular mechanism of EGFR for the diagnosis of NSCLC, the development of new drugs and the formulation of new treatment strategies to prolong the survival of patients.”

Q3) It has been shown recently that 20% of Lung cancer has RAS mutation (PMID: 33870211) while 22% of LAUD has KRAS mutation. Authors should discuss one of the future aspects of this study is RAS-mediated pathways in LAUD which might be involved in creating resistance. The discovery of KRAS G12C inhibitor resistance is one of the fields in lung cancer that is more concerned. The approach taken by this manuscript has some direct future implications in RAS therapy. The authors should add a few lines discussing this aspect. 

A3) Thank you for your comment. The potential impact of the research methods used in this manuscript on the treatment of RAS was discussed and presented in the 'Discussion' section.“In addition, the approach of this study provides a new idea to address tumor drug resistance. The research has indicated that RAS dysregulation and resulting signaling dysregulation account for one-third of all human cancers, and mutations in the RAS are usually related to treatment resistance and poor prognosis. As a key cancer driver, RAS has always been the focus of an intensive search for therapeutic approaches. So far, however, no effective RAS inhibitors have been approved for clinical use. Recently, the clinical results of KRAS G12C inhibitors have sparked excitement in the scientific community. Nevertheless, acquired drug resistance may limit the efficacy of inhibitors, indicating that combination therapy may be required. By analyzing RAS-related dysregulated genes and constructing a RAS-centered ceRNA network, oncogenic RAS and its downstream signaling and metabolic programs can be more effectively targeted. An accurate understanding of the coordinated interactions between RAS and other genes in the associated ceRNA network will be very important for developing novel targeted therapies for RAS-driven cancers.”

Once again, thank you very much for your constructive comments and suggestions which would help us both in English and in depth to improve the quality of the paper.

Kind regards.

Reviewer 2 Report

My main issues are with your data expression and analyses description.  All correlations examined should have a descriptor of r-squared (coefficient of determination) rather than r, so that the reader understands the proportion of the variance that may be explained by your experimental correlation analysis.  As you have done in your supplementary data document, please provide the 95% confidence interval around the correlations examined and presented in the main manuscript submission (grey shading).

All Figure legends should stand independently and have abbreviations used as part of the legend.  Also, the statistical approach needs to be provided in the legend to the figure.  

In the legends to the figures, the sample size is mostly not provided.  This needs to be part of the legend or part of the labelling of the relevant figure panels.

Some of your correlation analyses appear to have a nonlinear relationship.  Did you examine the data to determine whether or not a nonlinear fit would better explain the correlation examined?  If not why not?  Much of your data are near the origin of the XY plot.  Does this tend to bias the strength of the correlation, since much of the data with higher values of X are sparse and could bias the quality of the regression line.

In the end, the use of r-squared may require that you modify the strength of the associations you advocate in the manuscript.  This may require revision of the Results and Discussion sections.

Author Response

Dear professor:

I would like to thank you for your helpful review of my paper. Your comments are extremely useful, and I really appreciate your kindness. We have studied the comments carefully and made corrections which we hope meet with approval. The manuscript has modified according the comments.

Sincerely

Wen Li

2022.10.6

Responses to Reviewer

Q1) My main issues are with your data expression and analyses description.  All correlations examined should have a descriptor of r-squared (coefficient of determination) rather than r, so that the reader understands the proportion of the variance that may be explained by your experimental correlation analysis. As you have done in your supplementary data document, please provide the 95% confidence interval around the correlations examined and presented in the main manuscript submission (grey shading).

A1) Thank you for your comment. R indicates the strength of the linear correlation between two variables and is used to determine whether there is a linear correlation between them. In contrast, r-squared expresses the degree of contribution of the variable to the sum of squares of the total deviations, emphasising how good or bad the fit is between the models. We referred to methods in related papers (PMID: 29720189; PMID: 30207103; PMID: 30237435), as well as the GEPIA (http://gepia.cancer-pku.cn/detail.php) and TIMER (https://cistrome.shinyapps.io/timer/) databases, both of which used r to assess the correlation between the two sets of genes. Therefore, r was used to assess the strength of linear correlation between two genes in this study. In addition, we have demonstrated 95% confidence intervals (grey shading) around the correlations studied in the main manuscript.

Q2) All Figure legends should stand independently and have abbreviations used as part of the legend.  Also, the statistical approach needs to be provided in the legend to the figure.  

A2) Thank you for your comment. We have modified the legends so that all legends stand alone. The statistical methods used are also presented in the legends.

“Figure 1. High expression of EGFR in LUAD and prognostic value. (A) EGFR expression distribution in whole cancer tissues. (B) cBioPortal OncoPrint plot displaying the distribution of EGFR genomic alterations rate in LUAD dataset. (C) Validation of EGFR expression at the translational level by HPA database, normal tissue (1 & 2), tumor tissue (3 & 4). (D & E) EGFR expression in tumor- paracancer paired and tumor-normal samples. (F) The low-expression (n=243) and high-expression (n=244) of EGFR was compared by Kaplan-Meier survival curve. (G) Correlation of EGFR expression with TNM stage, stage I, n=272; stage II, n=119; stage III, n=84; stage IV, n=25. (H) Correlation between EGFR expression and tumour size, T1, n=166; T2-T4, n=337. (I) Correlation between EGFR expression and tumour lymph node metastasis, M0, n=339; M1, n=23. (J) Correlation between EGFR expression and distant metastases, N0, n=325; N1, n=93; N2-N3, n=76. The RNAs expression value was logarithmized with log10. Statistically significant differences between the two groups of data were estimated by Mann-Whitney test and independent t-test. One-way ANOVA was used to assess statistical differences between multiple groups of data. p < 0.05 was considered as statistically significant.”

“Figure 2. Volcano map of DERNAs between EGFRhigh and EGFRlow expression in LUAD samples and construction of ceRNA network. Red indicates upregulation, blue represents downregulation. (A) Volcano plots of differentially expressed lncRNAs (DElncRNA) (|log2FC| > 0.70, p < 0.05). (B) Volcano plots of differentially expressed miRNAs (DEmiRNA) (|log2FC| > 0.50, p < 0.05). (C) Volcano plots of differentially expressed mRNAs (DEmRNA) (|log2FC| > 0.50, p < 0.05). (D) The triple regulatory network in LUAD. (E) Functional enrichment analysis of mRNAs in the network. (F) Survival-related DERNAs regulatory network in LUAD.”

“Figure 3. Correlation analysis of differentially expressed RNAs in ceRNA network with overall survival of LUAD patients by Kaplan - Meier survival curve. (A) lncRNAs. (B) miRNAs. (C) mRNAs. Statistically significant differences between the two groups of data was estimated by Mann-Whitney test. p < 0.05 was considered as statistically significant.”

“Figure 4. Statistically significant differences between the two groups of data was estimated by independent t-test. p < 0.05 was considered as statistically significant.”

“Figure 6. Statistically significant differences between the two groups of data was estimated by independent t-test. p < 0.05 was considered as statistically significant.”

Q3) In the legends to the figures, the sample size is mostly not provided.  This needs to be part of the legend or part of the labelling of the relevant figure panels.

A3) Thank you for your comment. We have supplemented the sample size in the legend and associated images. Detailed information can be found in A2.

Q4) Some of your correlation analyses appear to have a nonlinear relationship. Did you examine the data to determine whether or not a nonlinear fit would better explain the correlation examined? If not why not? Much of your data are near the origin of the XY plot. Does this tend to bias the strength of the correlation, since much of the data with higher values of X are sparse and could bias the quality of the regression line.

A4) Thank you for your comment. In this study, we used linear fit to perform experimental correlation analysis. This is because we aim to find new target genes for the diagnosis and treatment of LUAD by analysing the EGFR-related ceRNA network. A necessary condition for ceRNA network composition is a positive correlation between lncRNA expression and mRNA expression, while both are negatively correlated with miRNA. Therefore, linear analysis can not only reflect the correlation strength between lncRNA, miRNA and mRNA, but also show the positive or negative correlation between groups. In this study, we analysed differentially expressed genes were analyzed in LUAD samples through the R package "edgeR", and the "Trimmed Mean of M-values" (TMM) normalization method was used to calibrate the read length of genes. The original characteristics of the data have been retained to the greatest extent possible. Although most of the data are relatively concentrated, there are still a small number of data with large differences. Although this may affect the quality of the regression line to some extent, the authenticity of the data was guaranteed to the greatest extent possible.

Q5) In the end, the use of r-squared may require that you modify the strength of the associations you advocate in the manuscript.  This may require revision of the Results and Discussion sections.

A5) Thank you for your comment. We have carefully considered your suggestion and have investigated it carefully. It was ultimately concluded that r might be more applicable to assess the strength of linear correlation between the two genes. Details of the specific analysis are given in response 1 (A1). We have therefore not made major changes to the relevant results and discussion sections.

Once again, thank you very much for your constructive comments and suggestions which would help us both in English and in depth to improve the quality of the paper.

Kind regards.

Round 2

Reviewer 1 Report

All concerns have been addressed, ready for acceptance. 

Author Response

Thank you very much for your support and recognition of our work. We wish you well in your work and a happy life!

Reviewer 2 Report

1. Generic drug names need not be capitalized in the text of the manuscript. - e.g. line 89.

2.  For your correlation analyses, I appreciate that the 95% confidence interval is now provided.  However, it is not defined as such in the relevant Figure legends.  This should be done where this type of analysis is provided.  Otherwise, it is left to the reader to guess what the shaded areas actually represent.

Author Response

Dear professor:

I would like to thank you for your helpful review of my paper. Your comments are extremely useful, and I really appreciate your kindness. We have studied the comments carefully and made corrections which we hope meet with approval. The manuscript has modified according the comments.

Sincerely

Wen Li

2022.10.7

Responses to Reviewer

Q1) Generic drug names need not be capitalized in the text of the manuscript. - e.g. line 89.

A1) Thank you for your comment. We have amended the name of the drug in line 89. “Although EGFR tyrosine kinase inhibitors (EGFR-TKIs), including erlotinib and gefitinib, have shown initial efficacy in 30% of NSCLC patients with EGFR mutations in the past few decades, secondary resistance often occurs in EGFR-TKIs treatment.”

Q2) For your correlation analyses, I appreciate that the 95% confidence interval is now provided.  However, it is not defined as such in the relevant Figure legends.  This should be done where this type of analysis is provided. Otherwise, it is left to the reader to guess what the shaded areas actually represent.

A2) Thank you for your comment. We have modified the legend to Figure 5 by adding an illustration of the confidence intervals. “(A) Correlation analysis between LINC00460, mir-338-3p and MCM4 in LUAD (n=515). Grey shaded areas indicate 95% confidence intervals.”

Once again, thank you very much for your constructive comments and suggestions which would help us both in English and in depth to improve the quality of the paper.

Kind regards.
